# Protective Effect of *Ganoderma atrum* Polysaccharide on Acrolein-Induced Apoptosis and Autophagic Flux in IEC-6 Cells

**DOI:** 10.3390/foods11020240

**Published:** 2022-01-17

**Authors:** Yudan Wang, Xinxin Chang, Bing Zheng, Yi Chen, Jianhua Xie, Jialuo Shan, Xiaoyi Hu, Xiaomeng Ding, Xiaobo Hu, Qiang Yu

**Affiliations:** State Key Laboratory of Food Science and Technology, China-Canada Joint Laboratory of Food Science and Technology (Nanchang), Key Laboratory of Bioactive Polysaccharides of Jiangxi Province, Nanchang University, 235 Nanjing East Road, Nanchang 330047, China; W15698331381@163.com (Y.W.); cxx13870698052@163.com (X.C.); ncuspyzhengbing@163.com (B.Z.); chenyi15@ncu.edu.cn (Y.C.); jhxie@ncu.edu.cn (J.X.); jialuoa1998@163.com (J.S.); huxiaoyi0101@126.com (X.H.); dxm19970128@163.com (X.D.); hxbxq2005@163.com (X.H.)

**Keywords:** polysaccharide, acrolein, IEC-6 cells, autophagy, apoptosis

## Abstract

This study was designed to explore the beneficial effect and mechanism of *Ganoderma atrum* (G. atrum) polysaccharide (PSG-1) on acrolein-induced IEC-6 cells. Our results indicated that PSG-1 significantly reduced the impairment of acrolein on cell viability, decreased oxidative stress, and enabled normal expression of tight junction (TJ) proteins that were inhibited by acrolein in IEC-6 cells. Furthermore, PSG-1 attenuated the elevation of microtubule-associated proteins light chain 3 (LC3) and Beclin 1-like protein 1 (Beclin 1) and increased the protein levels of phospho-mTOR (p-mTOR) and phospho-akt (p-akt), indicating that PSG-1 activated the mammalian target of rapamycin (mTOR) signaling pathway and alleviated acrolein-induced autophagy in IEC-6 cells. Moreover, PSG-1 markedly attenuated the acrolein-induced apoptosis, as evidenced by the increase in mitochondrial membrane potential (MMP) and B-cell lymphoma 2 (Bcl-2) expression, and the decrease in cysteine aspartate lyase (caspase)-3 and caspase-9. In addition, autophagy the inhibitor inhibited acrolein-induced TJ and apoptosis of IEC-6 cells, while the apoptosis inhibitor also inhibited acrolein-induced TJ and autophagy, suggesting that autophagy and apoptosis were mutually regulated. Taken together, the present study proved that PSG-1 could protect IEC-6 cells from acrolein-induced oxidative stress and could repair TJ by inhibiting apoptosis and autophagic flux, where autophagy and apoptosis were mutually regulated.

## 1. Introduction

Acrolein is an electrophilic, unsaturated aldehyde, a colorless, flammable, volatile, pure liquid, known for its pungent odor and strong irritation of mucous membranes [1], which comes from food, the environment, and the human body itself [2,3]. Acrolein in food is mainly produced by oil fumes or fried and baked foods, which produce acrolein mainly through the Maillard reaction and lipid oxidation [4]. Acrolein is neurotoxic, genotoxic, and potentially carcinogenic, and the question of how to mitigate the side effects of acrolein has generated scholars’ attention [5]. Therefore, finding active substances that can reduce or antagonize the toxic effects of acrolein in food is important to protect humans from its harm.

Programmed cell death (PCD) is a self-regulatory mechanism of the organism, mainly including autophagy, apoptosis, and necrosis, the first two of which can play an indispensable role in the life activities of the organism by working closely together. Autophagy is a highly conserved intracellular degradation pathway, a protective mechanism for cells in response to various of hazardous stimuli, which maintains intracellular homeostasis by degrading damaged macromolecular proteins or organelles within the cell [6]. Autophagy is a multi-step regulated process with highly complex signaling. In this process, microtubule-associated proteins light chain 3 (LC3), Beclin 1-like protein 1 (Beclin 1), and so on, play remarkably important roles [7]. Apoptosis is an orderly and autogenous cell death that is regulated by the involvement of genes, acting in the extrinsic and intrinsic pathways [8]. Both pathways stimulate cysteine aspartate lyase (caspase), which is the basis of apoptosis [9]. The endogenous pathway surrounding mitochondria is mainly mediated by the proteins of B-cell lymphoma 2 (Bcl-2) family. Several structural features are altered during apoptosis: membrane blistering, cytoplasmic crumpling, DNA breaks, chromatin condensation, and DNA degradation of chromosomes [10].

The intestine is the body’s largest immune system, which mainly relies on the intestinal mucosal barrier to resist the invasion of foreign toxic substances and prevent the invasion of adverse substances and displacement in the intestine [11]. Thus, the integrity of the intestinal mucosal screen is essential to maintain the normal physiological function of the intestine [12]. Many studies have found that autophagy and apoptosis act as vital players in the intestinal barrier and work together to sustain the complexity of the intestinal barrier [13].

Numerous studies have revealed that polysaccharides possessed great promises for food and health applications [14,15,16]. *Ganoderma atrum* (*G. atrum*) has been used as a medicine or diet with high nutritional value for thousands of years [17]. In the last decade, our laboratory has identified a polysaccharide from *G. atrum* (PSG-1), with purity > 99.8%. It is composed of glucose (Glc), mannose (Man), galactose (Gal), and galacturonic acid (GalA) in molar ratio of 4.91:1:1.28:0.71, and is rich in 17 proteins, including glutamic acid, asparagic acid, alanine, glycine, threonine, and serine; in addition, it is formed as a main chain of 1,3-linked and 1,6-linked β-Glcp residues, substituted at O-3 and O-6 position as the branch points, with the molecular weight of 1013 kDa. The residues of α-1,4-Galp, α-1,2-Manp, and α-1,4-Manp were also found in the backbone. Side chains were terminated by β-Glcp, with the composition of α-1,4-Galp, α-1,4-GalpA, β-1,3-Glcp, and β-1,6-Glcp [18]. PSG-1 have exhibited a broad range of beneficial health effects, including immunomodulatory, chemo-protective, antioxidant, and hypoglycemic activities [19,20,21,22]. In particular, latest studies have also proved that PSG-1 could modulate the mucosal immunity through epithelial cells [23,24]. However, there are few reports on acrolein-induced intestinal effects. Therefore, in this study, an acrolein-induced IEC-6 cells injury model was established, aiming to examine the protective characteristics of PSG-1 against acrolein-induced intestinal injury and its intrinsic mechanism.

## 2. Materials and Methods

### 2.1. Materials

Acrolein was purchased from Shandong Xiya Chemical Industry Co., Ltd. (Linyi, China). 3-Methyladenine (3-MA) (PI3K inhibitor) and Z-DEVD-FMK (caspase-3 inhibitor) were purchased from MedChem Express Co., Ltd. (Monmouth Junction, NJ, USA). Annexin V-FITC apoptosis detection kits were purchased from Beyotime Biotechnology (Nanjing, China). Mitochondrial membrane potential (JC-1) assay kit was obtained from Solarbio Science and Technology Co., Ltd. (Beijing, China). The Cell Counting Kit-8 (CCK-8) was from Dojindo Molecular Technologies, Inc. (Shanghai, China).

### 2.2. Cell Culture

IEC-6 cells (rat-derived intestinal epithelial cell line) were cultured in high glucose Dulbecco’s Minimal Essential Medium (DMEM) (Solarbio Science and Technology Co., Ltd. Beijing, China), supplemented with 10% fetal bovine serum (FBS) (BI, 04-001-1ACS), 5% CO_2_, temperature 37 °C and humidity saturation conditions. The passages 5–20 were used in this study.

### 2.3. Cell Viability

A 96-well microplate format was used. After 24 h of incubation, treating IEC-6 cells with acrolein (0–80 μM) for 20 h, a solution of CCK-8 was added to each well. Finally, the optical absorbance was read at 450 nm with a microplate reader (Thermo Fisher Scientific, Waltham, MA, USA). Under the same cultural conditions as above, PSG-1 at concentrations of 20, 40, 80, and 160 (μg/mL) were added to each group, while a blank control group was set up. The effect of PSG-1 on IEC-6 cells viability was detected.

### 2.4. Antioxidant Enzyme Assays

After cell culture, cells were washed once with phosphate-buffered saline (PBS), cells were digested with trypsin, and they were collected by centrifugation for 10 min at 1000× *g*. Protein concentrations were determined using the BCA protein assay kit, superoxide dismutase (SOD), malondialdehyde (MDA), and glutathione peroxidase (GPx) (Beyotime Biotechnology Nanjing, China) were estimated by referring to the instructions.

### 2.5. Apoptosis Rate Detection

The IEC-6 cells were collected after washing with PBS and digesting with trypsin, centrifuged, and 195 μL binding buffer was added to keep the cells in suspension; 5 μL annexin V-FITC and 10 μL PI solutions were added and mixed. The cells were then cultured at 20–25 °C for 10–20 min, protected from light, observed by flow cytometry, and detected within 1 h.

### 2.6. Assay for Mitochondrial Membrane Potential (MMP)

Cells in 6-well plates were washed once with PBS, then 1 mL of cell culture solution and 1 mL of JC-1 staining buffer were added and mixed thoroughly. Cells were cultured in an incubator at 37 °C for 20 min. Following staining, the cells were polished twice with JC-1 staining buffer and observed by fluorescence microscopy.

### 2.7. Western Blot Analysis

Protein samples were prepared by treating IEC-6 cells with acrolein and PSG-1 in 6-well plates, after 10% sodium dodecyl sulfate (SDS)-PAGE electrophoresis, and transported to a PVDF membrane (Millipore Co., Belford, MA, USA). The membranes were blocked with 5% bovine serum albumin (BSA) for 1 h. Membrane were incubated with primary antibodies (1:1000): ZO-1, claudin-1, occludin, LC3B, Beclin 1, phospho-mammalian target of rapamycin (p-mTOR), phospho-Protein Kinase B (p-akt), caspase-3, caspase-9, Bcl-2, and beta actin monoclonal antibody (β-actin) (CST, Boston, MA, USA) at 4 °C overnight under shaking and incubated with horseradish peroxidase-conjugated secondary antibodies (ZSGB Biotechnology, Beijing, China) for 1 h at room temperature. After incubation in ECL detection reagent, the fluorescent signal was detected using the Molecular Imager ChemiDoc™ XRS Imaging System (Bio-Rad Laboratories, Hercules, CA, USA).

### 2.8. Inhibitor Experiments

According to the instructions, cells induced by acrolein were pretreated with configured autophagy inhibitor 3-MA and the apoptosis inhibitor Z-DEVD-FMK, respectively. Cellular proteins were extracted after a certain period of time, and the proteins expression were examined by Western blot method to explore the interaction between TJ and autophagy, TJ and apoptosis, and the autophagy and apoptosis pathways.

### 2.9. Statistical Analysis

Statistical analysis was carried out using Graphpad Prism 6.01 software. Results were analyzed by a one-way ANOVA analysis of variance and expressed as the mean ± SD. *p* < 0.05 was considered statistically significant.

## 3. Results

### 3.1. PSG-1 Increased the IEC-6 Cells Viability Exposed to Acrolein

As shown in Figure 1A, acrolein inhibited the viability of IEC-6 cells in a dose-dependent manner. Besides, when acrolein concentration was 40 μM, the cell viability was about 50%. Thus, we chose an acrolein concentration of 40 μM for subsequent experiments to detect the potential cytoprotective capacity of PSG-1 (*p* < 0.01).

Acrolein treatment caused a significant decrease in cell viability, while different concentrations of PSG-1 significantly inhibited the toxicity of acrolein. As can be seen in Figure 1B, all 4 concentration groups were effective, and cell viability tended to enhance and then diminish with increasing PSG-1 concentration, especially at a dose of 80 μg/mL, which almost restored cell viability to normal levels. The differences between the groups were not significant (*p* < 0.01).

### 3.2. PSG-1 Attenuated Acrolein-Induced Oxidative Damage of IEC-6 Cells

As shown in Figure 2A, the cellular SOD activity declined after acrolein treatment, the addition of PSG-1 restored the SOD activity to different degrees, and the PSG-1 dose group with 160 μg/mL had the highest effect on the increase in SOD activity.

In Figure 2B, the MDA level significantly increased after the addition of acrolein, while PSG-1 treatment significantly decreased the MDA contents.

The significant decrease in GPx viability was caused by acrolein. Under the protection of PSG-1, GPx significantly increased especially in the 160 μg/mL dose group (Figure 2C), indicating that PSG-1 dramatically prevented the inhibition of acrolein-induced oxidative stress (*p* < 0.05).

### 3.3. Effects of PSG-1 on TJ Proteins Damaged by Acrolein in IEC-6 Cells

TJ mainly includes transmembrane proteins and intracellular plaque proteins, such as Zonula occludens (ZO), different claudin-1 family members, and occludin. As Figure 3 shows, acrolein-induced expression of ZO-1, claudin-1, and occludin were strikingly reduced, indicating that acrolein treatment significantly impaired intestinal barrier function, and PSG-1 treatment clearly promoted the expression of the three proteins (*p* < 0.05), suggesting that PSG-1 was able to tightly link the reduction in proteins and thus protect the intestinal barrier.

### 3.4. Effects of PSG-1 on Autophagic Proteins Damaged by Acrolein in IEC-6 Cells

Figure 4B results revealed that the LC3-II/I ratio was significantly higher in acrolein-treated cells, while LC3 protein was significantly lower in PSG-1-treated cells than the model group. In addition, we examined key proteins of the PI3K/Akt/mTOR pathway and the Beclin 1 pathway, the phosphorylation levels of mTOR and akt were clearly lower and the level of Beclin 1 was significantly higher in acrolein-induced cells than in the control group. After treating IEC-6 cells with PSG-1, the levels of p-mTOR and p-akt were significantly enhanced, and the content of Beclin 1 was significantly lower than that of the acrolein group (*p* < 0.05) (Figure 4C–E).

### 3.5. Effect of PSG-1 on Acrolein-Induced Apoptosis

Double staining with annexin V and PI to detect apoptosis showed that the left lower quadrant had live cells and the right lower quadrant had apoptotic cells. Figure 5 showed that after acrolein induction, the apoptosis rate of cells increased significantly, reaching more than 70%, far exceeding that of normal controls. A decrease in the percentage of apoptotic cells could be clearly observed after PSG-1 treatment.

### 3.6. Effects of PSG-1 on MMP Induced by Acrolein in IEC-6 Cells

The process of apoptosis is often accompanied by a disruption of the MMP, which is thought to be one of the earliest events in the apoptotic process. The change of MMP was detected by JC-1 fluorescent probe. It can be seen that cells after acrolein induction have depolarized mitochondrial transmembrane potential and impaired mitochondrial permeability, while PSG-1 treatment restored cell morphology with enhanced red fluorescence intensity and weaker green fluorescence intensity, implying that PSG-1 attenuated acrolein-induced loss of MMP in IEC-6 cells, thereby preserving mitochondria (Figure 6).

### 3.7. Effects of PSG-1 on Apoptotic Proteins Damaged by Acrolein in IEC-6 Cells

As depicted in Figure 7, acrolein significantly promoted the two positively related proteins, caspase-3 and caspase-9, and significantly declined the level of anti-apoptotic protein Bcl-2. The addition of PSG-1 decreased the expression of caspase-3 and caspase-9 proteins and increased the content of Bcl-2 protein (*p* < 0.05), which can indicate that PSG-1 can inhibit acrolein-induced apoptosis.

### 3.8. Effects of Autophagy Inhibitor and Apoptosis Inhibitor on the TJ in IEC-6 Cells

After treating cells with autophagy inhibitor and apoptosis inhibitor, we separately inspected the expression of three TJ proteins, and it was found that the expression of TJ protein was markedly increased in IEC-6 cells, where the inhibitor co-interacted with acrolein, compared with acrolein alone (*p* < 0.05) (Figure 8).

### 3.9. The Effects of Autophagy Inhibitor on IEC-6 Cell Apoptosis

Autophagy inhibitor treatment of IEC-6 cells significantly decreased the increase in acrolein-induced autophagy-related protein LC3 (Figure 9B). After that, we examined the expression of apoptosis-related proteins Bcl-2, caspase-3, and caspase-9, and the results demonstrated that the autophagy inhibitor reduced the overexpression of acrolein-induced, apoptosis-related proteins caspase-3 and caspase-9 and increased the expression of anti-apoptotic protein Bcl-2 (*p* < 0.05) (Figure 9C–E).

### 3.10. The Effects of Apoptosis Inhibitor on IEC-6 Cell Autophagy

In Figure 10B, treatment of IEC-6 cells with apoptosis inhibitors markedly inhibited the increase in acrolein-induced, apoptosis-related protein caspase-3. To verify the apoptosis inhibitor effect on cellular autophagy, we examined the expression of autophagy-related proteins Beclin 1, and the result showed that apoptosis inhibitor significantly reduced acrolein-induced overexpression of autophagy-related protein Beclin 1 (*p* < 0.05) (Figure 10C).

## 4. Discussion

The intestinal epithelial cells (IECs) have roles in the functions of digestion, absorption of nutrients [25], and the barrier function preventing toxic substances from entering the body, etc. [3,26]. They are the main component of the mechanical barrier of the intestinal mucosa and the first line of defense of the body against intestinal bacteria and toxins [27]. In this study, acrolein was found to cause a reduction in growth and viability of IEC-6 cells, and its cytotoxic action on IEC-6 cells was a concentration-dependent action, with a concentration of 40 μM significantly impairing the viability. Meanwhile, several concentrations of PSG-1 extracts were found to have cytoprotective effects against acrolein-induced damage in IEC-6 cells.

GPx is an essential peroxidase enzyme that is extensively present in living organisms [28]. MDA and SOD content is a vital indicator of the potential antioxidant capacity of the body [29,30]. Meanwhile, we found that acrolein induced oxidative stress in IEC-6 cells, decreased the SOD activity and GPx activity, and increased the intracellular lipid peroxide MDA content. PSG-1 prevented acrolein-induced oxidative stress by improving GPx and SOD levels and reducing MDA content. Therefore, we hypothesized that PSG-1 altered the antioxidant activity in IEC-6 cells under the effect of acrolein to enhance the antioxidant capacity of cells and prevented oxidative stress damage to cells, which indirectly achieved cell protection. The mechanism may be due to the PSG-1 combining with radicals to form a stable radical, or combining with the radical ions, which are necessary for radical chain reaction, and then the reaction is terminated [18].

The TJ of IECs is the main component connecting IECs to the intestinal mucosal barrier, which regulates the permeability of barrier and is the first line of defense of cell bypass [31]. The TJ determines the electrolyte and water balance in the intestine, which is the key factor of intestinal integrity and mucosal barrier function, and it consists mainly of occludin, claudins, and other protein families, which are composed of the linker complex proteins (ZO-1), and cytoskeletal structures which form the TJ complex [32]. Three common TJ proteins—occludin, claudin, and ZO-1—were assayed. Acrolein was able to decrease the level of TJ proteins, while the PSG-1 treatment markedly enhanced the expression of these proteins, which is beneficial for maintaining the integrity of the intestinal mucosa. This is consistent with the oxidative damage index data. The experiment suggests that PSG-1 can maintain intestinal barrier function by up-regulating the expression level of TJ proteins in IEC-6 cells.

Autophagy, known as type II programmed cell death, is a self-protective mechanism widely present in eukaryotic cells [33]. To confirm autophagy is engaged in the protection of acrolein-damaged IEC-6 cells by PSG-1, the expression of key proteins of autophagy was inspected. The expression ratios of LC3-II and LC3-I can be used as important indicators for autophagy experiments. The ratio of LC3-II/I was significantly higher in IEC-6 cells treated with acrolein, suggesting that autophagy was promoted by acrolein exposure.

Autophagy is a complex signaling process with multi-step regulations. Studies have revealed that autophagy mainly consists of the PI3K/Akt/mTOR pathway and the Beclin 1 pathway, which is a key regulation protein of autophagy [34]. The mTOR signaling pathway is a critical regulation of autophagy initiation, activated mTOR inhibits autophagy [35]. After acrolein treatment, the expression of Beclin 1 increased, and the p-mTOR and p-akt content significantly decreased. Moreover, PSG-1 therapy significantly reduced LC3-II/I and Beclin 1 expression and promoted p-mTOR and p-akt expression. Experiments suggested that PSG-1 was able to inhibit acrolein-stimulated excessive autophagy, which may be achieved by promoting the mTOR pathway.

Apoptosis is a fundamental and important biological phenomenon that plays an essential role in the clearance of abnormal cells [36]. It was found that acrolein caused apoptosis of IEC-6 cells at pathological concentrations. To investigate whether apoptosis has a function in acrolein-induced IEC-6 cells, we first looked at the rate of apoptotic cell death. The experiments suggested that the apoptosis rate of cells remarkably elevated after acrolein induction and decreased significantly after the addition of PSG-1, which suggested that acrolein can induce IEC-6 cells apoptosis and PSG-1 can protect IEC-6 cells by reducing the number of apoptotic cells. Various studies have shown that mitochondria are closely related to apoptosis and that apoptosis is irreversible once the transmembrane potential of the mitochondrial membrane collapses. Therefore, the change of MMP was detected, the result showed that IEC-6 cells treatment with acrolein caused MMP loss and PSG-1 treatment significantly improved this situation.

Afterwards, the expression of apoptosis-related proteins was also examined. Bcl-2 is located upstream of mitochondria and an important regulation of the altered permeability of mitochondrial membranes. Caspase-3 and caspase-9 play an irreplaceable role in apoptosis. Caspase-9 involves the initiation of apoptosis. Caspase-3 is the most critical apoptosis executor downstream of the caspase cascade reaction. We found that after acrolein induction of IEC-6 cells, the expression of Bcl-2 was significantly decreased, the expression of two apoptotic proteins, caspase3 and caspase-9, was significantly increased, and PSG-1 restored the expression of these three proteins to a certain extent. The above data suggested that acrolein-induced IEC-6 cells led to MMP loss, increased expression of caspase 3 and caspase 9 and decreased expression of the Bcl-2 protein, which indicated that PSG-1 effectively restored acrolein-induced apoptosis through the mitochondrial pathway. It had been established that the biological activity of mushroom polysaccharides were attributed to their chemical structure [14,37]. Thus, we speculated that the regulation of autophagy and apoptosis by PSG-1 was most likely related to the main structure β-(1 → 3, 1 → 6)-glucan, which was elucidated in our previous study [18,38].

Numerous studies have shown that autophagy and apoptosis play an invaluable function in the intestinal barrier and jointly maintain the integrity of the intestinal barrier. To explore the association between autophagy/apoptosis and TJ, we pretreated IEC-6 cells with specific inhibitors of autophagy and apoptosis pathways, respectively, examining the expression of occludin, claudin, and ZO-1 in each group. It was found that the expression of TJ proteins was increased after treatment of IEC-6 cells with two inhibitors, suggesting that acrolein-induced autophagy and apoptosis occur in IEC-6 cells could disrupt the integrity of the intestinal barrier.

It is well known that a complex interactive regulation exists between autophagy and apoptosis [39]. The relationship between autophagy and apoptosis in acrolein-induced IEC-6 cells by pathway inhibitors was explored in this experiment. It was revealed that the expression of autophagy-related proteins was reduced in autophagy inhibitor-treated IEC-6 cells, and that autophagy inhibitor inhibited the expression of key apoptotic proteins, suggesting that autophagy may be modulated by acrolein-induced apoptosis of IEC-6 cells. Apoptosis inhibitor-treated cells showed reduced expression of apoptosis-related proteins. In addition, the apoptosis inhibitor inhibited the expression of key autophagic proteins, suggesting that apoptosis may be modulated by acrolein-induced autophagy of IEC-6 cells.

## 5. Conclusions

This study proved that PSG-1 repaired oxidative stress in acrolein-induced IEC-6 cells, restored the expression of TJ proteins inhibited by acrolein, and attenuated acrolein-induced autophagy and apoptosis in IEC-6 cells. Furthermore, PSG-1 restored TJ by inhibiting apoptosis and autophagic flux, where autophagy and apoptosis were mutually regulated. The finding may provide a theoretical basis for the development and generation of mushroom β-(1 → 3, 1 → 6)-glucan as high value-added products.

## Figures and Tables

**Figure 1 foods-11-00240-f001:**
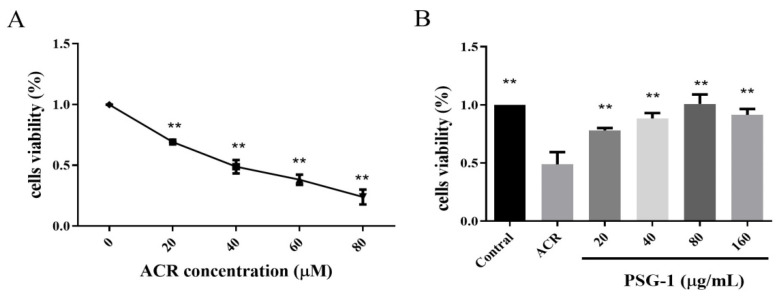
PSG-1 increased cell viability exposed to acrolein. (**A**) Acrolein effect on IEC-6 cells viability. (**B**) Effect of PSG-1 on acrolein-induced IEC-6 cells viability. Values are means ± SD (*n* = 3), ** *p* < 0.01 compared with the acrolein group.

**Figure 2 foods-11-00240-f002:**
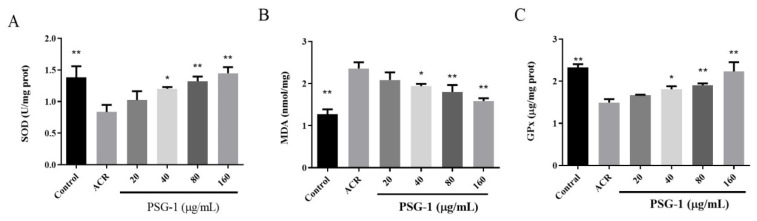
PSG-1 regulated acrolein-induced SOD, MDA, and Gpx. The oxidative state of IEC-6 cells, following acrolein (40 μM) and PSG-1 (20, 40, 80, and 160 μg/mL) treatment, was determined by measuring SOD (**A**), MDA (**B**) and GPx (**C**).Values are means ± SD (*n* = 3), * *p* < 0.05, ** *p* < 0.01 versus the acrolein group.

**Figure 3 foods-11-00240-f003:**
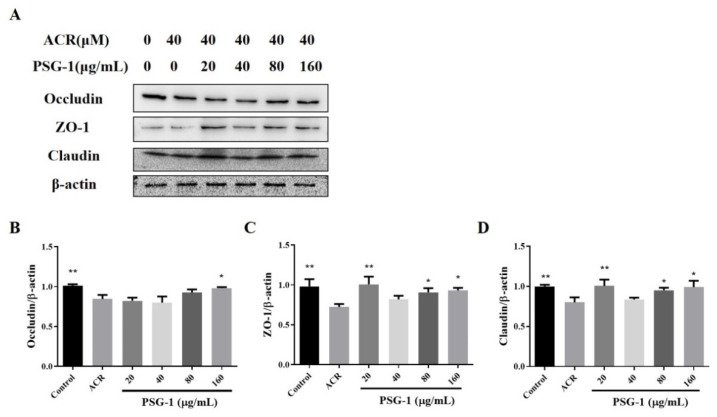
PSG-1 regulated the expression of TJ-related proteins in acrolein-induced IEC-6 cells. (**A**) The protein levels of occludin, claudin, and ZO-1. (**B**–**D**) Relative intensities of these proteins. Values are means ± SD (*n* = 3), * *p* < 0.05, ** *p* < 0.01 compared with the acrolein group.

**Figure 4 foods-11-00240-f004:**
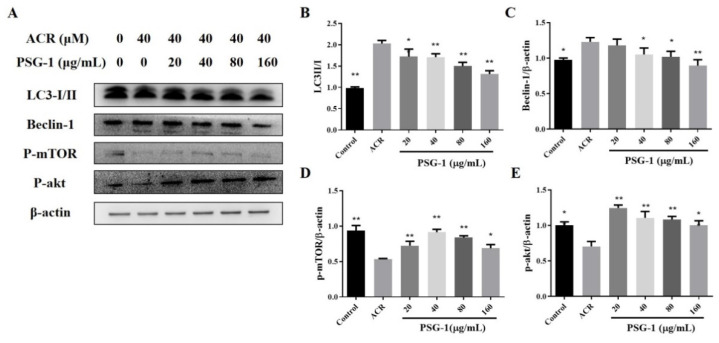
PSG-1-regulated, acrolein-induced autophagy-related proteins. (**A**) The protein levels of LC3, Beclin 1, p-mTOR, and p-akt. (**B**–**E**) Relative intensities of these proteins. Values are means ± SD (*n* = 3), * *p* < 0.05, ** *p* < 0.01 compared with the acrolein group.

**Figure 5 foods-11-00240-f005:**
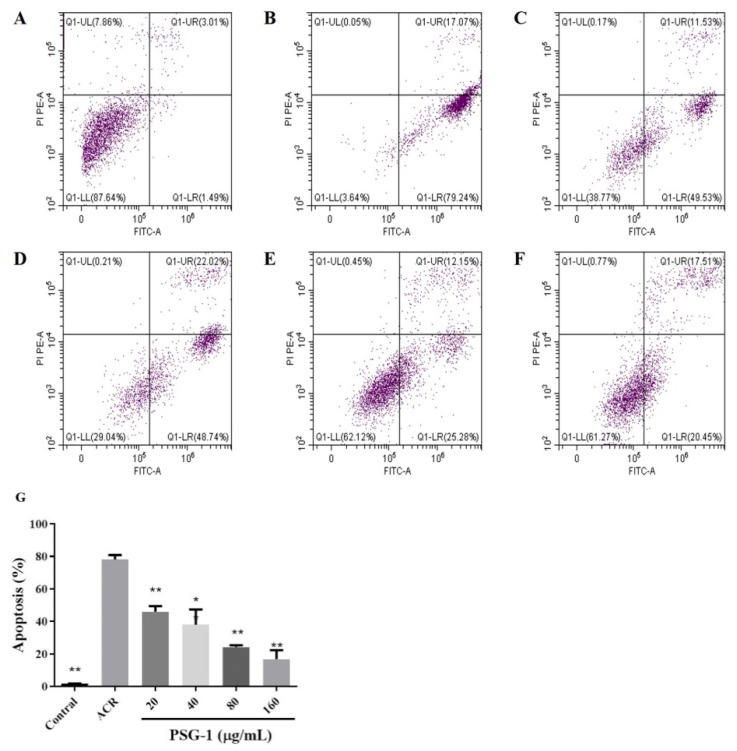
PSG-1 attenuated acrolein-induced apoptosis. Apoptosis measured by flow cytometry. (**A**) control cells; (**B**) the cells treated with acrolein, (**C**–**F**) the cells treated with acrolein and PSG-1 at concentrations of 20, 40, 80and 160μg/mL, (**G**) Effect of PSG-1 on acrolein-induced apoptosis. Values are means ± SD (*n* = 3), * *p* < 0.05, ** *p* < 0.01 compared with the acrolein group.

**Figure 6 foods-11-00240-f006:**
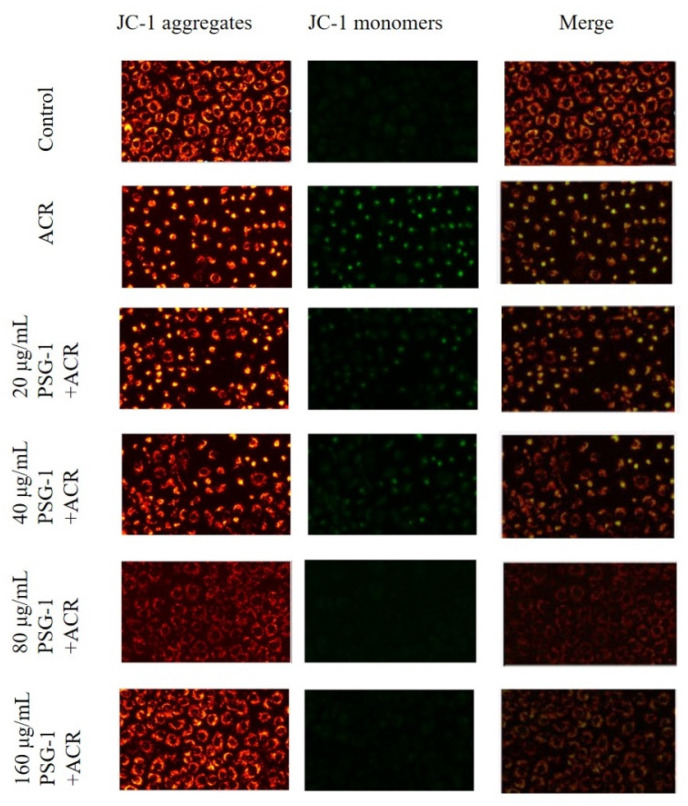
PSG-1 attenuated acrolein-induced MMP reduction. MMP was detected by fluorescence microscopy.

**Figure 7 foods-11-00240-f007:**
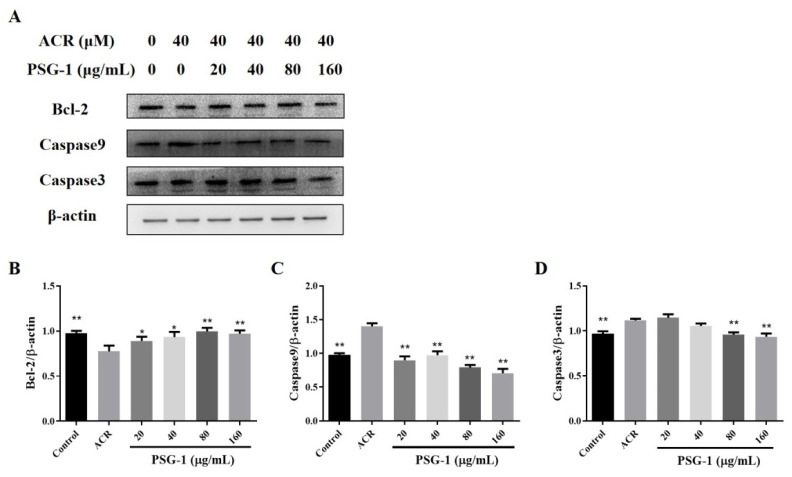
PSG-1 regulated the expression of apoptosis-related proteins in acrolein-induced IEC-6 cells. (**A**) The protein levels of caspase-9, caspase-3, and Bcl-2. (**B**–**D**) Relative intensities of these proteins. Values are means ± SD (*n* = 3), * *p* < 0.05, ** *p* < 0.01 compared with the acrolein group.

**Figure 8 foods-11-00240-f008:**
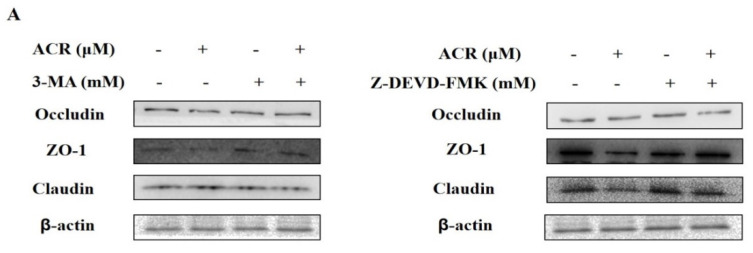
Autophagy and apoptosis inhibitor attenuated TJ. (**A**) The protein levels of occludin, claudin, and ZO-1, “+” indicates addition of acrolein or PSG-1 and “-” means not added. (**B**–**G**) Relative intensities of these proteins. Values are means ± SD (*n* = 3), * *p* < 0.05, ** *p* < 0.01 compared with the acrolein group.

**Figure 9 foods-11-00240-f009:**
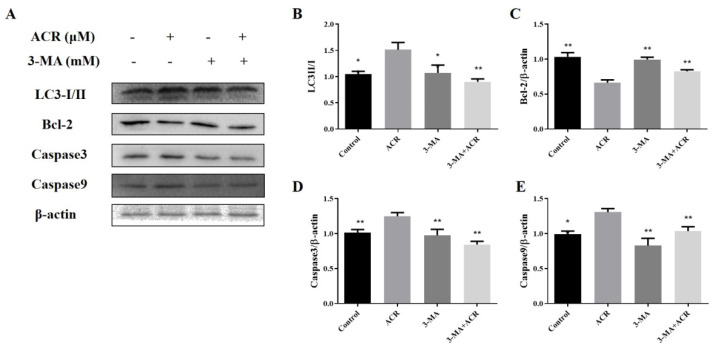
Autophagy inhibitor attenuated apoptosis. (**A**) The protein levels of LC3, caspase-9, caspase-3, and Bcl-2. (**B**–**E**) Relative intensities of these proteins. Values are means ± SD (*n* = 3), * *p* < 0.05, ** *p* < 0.01 compared with the acrolein group.

**Figure 10 foods-11-00240-f010:**
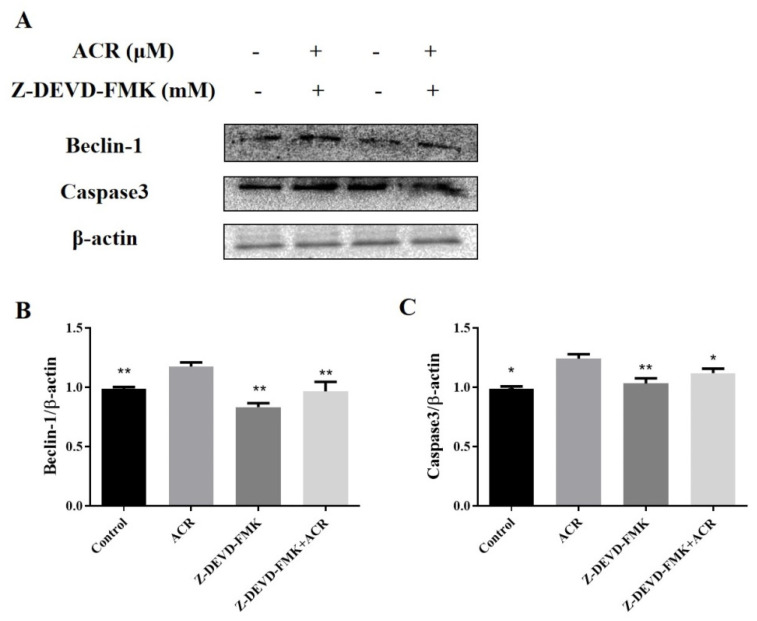
Apoptosis inhibitor attenuated autophagy. (**A**) The protein levels of caspase-3 and Beclin 1 proteins. (**B**,**C**) Relative intensities of these proteins. Values are means ± SD (*n* = 3), * *p* < 0.05, ** *p* < 0.01 compared to ACR group.

## Data Availability

Not applicable.

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
