# Peer review of "Protective Effect of Ganoderma atrum Polysaccharide on Acrolein-Induced Apoptosis and Autophagic Flux in IEC-6 Cells"

_foods, 2022, doi:10.3390/foods11020240_

Round 1

Reviewer 1 Report

The  manuscript concerns the study on the effect of  Ganoderma atrum-derived polysaccharide on acrolein-induced IEC-6 cells.

The toxic effects of acrolein (which is formed from the fats in fried and baked foods) are a serious problem. The search for food ingredients reducing these effects is fully justified. Thus, the research on the polysaccharide obtained by the Authors from Ganoderma atrum (an edible and medicinal mushroom) is interesting.

The  manuscript is the one of the series of the papers by the Authors, concerning the biological activity of the polysaccharide with the acronym PSG-1.

In general,  the methodology of the research, presentation of the results and conclusions  are correct and presented clearly.

However, I suggest several additions and changes to the manuscript:

- all abbreviations should be expanded the first time they appear in the text. The Authors do not follow this rule.

- for the Authors, the PSG-1 is a well-known structure  described in previous papers. For the reader - not necessarily. Although the Authors provide literature references, I believe that in the Discussion section they should precisely define the structure of the studied compound.

- biological activity of chemical compounds (including polysaccharides) is always a function of their structure. When analyzing the activity and mechanism of action of PSG-1 on acrolein-induced IEC-6 cells, the Authors should relate the results to the structure of the compound.

Mushroom polysaccharides are highly diversified (different structures - primary, secondary tertiary, etc.) so the Discussion and Conclusion proposed by the Authors should be addressed to a specific group of polysaccharide derivatives.

- Figures 2, 3 and 4 in the printed version of the manuscript appear to be of low resolution (not legible)

Author Response

Thank you very much for your good suggestion. We have considered your comments very carefully and made the revisions in the revised manuscript.

Comment 1: All abbreviations should be expanded the first time they appear in the text. The Authors do not follow this rule.

Response 1: Thank you very much for your good suggestion. According to your guideline, we have checked and revised in the revised manuscript, please read Pages 1-3 in the red font.

Comment 2: For the Authors, the PSG-1 is a well-known structure described in previous papers. For the reader-not necessarily. Although the Authors provide literature references, I believe that in the Discussion section they should precisely define the structure of the studied compound.

Biological activity of chemical compounds (including polysaccharides) is always a function of their structure. When analyzing the activity and mechanism of action of PSG-1 on acrolein-induced IEC-6 cells, the Authors should relate the results to the structure of the compound.

Mushroom polysaccharides are highly diversified (different structures - primary, secondary tertiary, etc.) so the Discussion and Conclusion proposed by the Authors should be addressed to a specific group of polysaccharide derivatives.

Response 2: Thank you very much for your good suggestion. Based on your guidance, we have added the description of structure information of PSG-1 in the introduction, and extended the discussion about activity and mechanism of action of PSG-1 on acrolein-induced IEC-6 cells. Please read Page2, 10-11 in the red font.

Comment 3: Figures 2, 3 and 4 in the printed version of the manuscript appear to be of low resolution (not legible).

Response 3: Thank you very much for your good suggestion. According to your guideline, we have uploaded the original image of Figure 2, 3 and 4.

Once again, thank you very much for your comments and suggestions. We really hope that you could be satisfied with our modifications and explanations. We are also willing to make further improvements if needed and thank you again for your work and time.

Reviewer 2 Report

The manuscript describes the protective effect of PSG-1polysaccharide of Ganoderma atrum on acrolein-induced IEC-6 cells.

The manuscript is well written with logical sequence, supported by results and discussions.

My only suggestion is the conclusions section which should be more extended.

Author Response

Thank you very much for your great support. We have considered your comments very carefully and made the revisions in the revised manuscript.

Reviewer 3 Report

This study investigated exploration of the beneficial effect and mechanism of Ganoderma atrum (G. atrum) polysaccharide (PSG-1) on acrolein-induced IEC-6 cells. As the authors claim, the results of this study prove that PSG-1 could protect IEC-6 cells from acrolein-induced oxidative stress, and repair TJ by inhibiting apoptosis and autophagic flux, where autophagy and apoptosis were mutually regulated. Therefore, I recommend this manuscript should make the following minor revisions.

  1. Some figures are unclear, whose resolutions should be improved.
  2. Conclusion should be revised as it concisely mentions entire findings through this study.
  3. English throughout the text should be revised.

Author Response

Thank you very much for your good suggestion. We have considered your comments very carefully and made the revisions in the revised manuscript.

Comment 1: Some figures are unclear, whose resolutions should be improved.

Response 1: Thank you very much for your good suggestion. According to your guideline, we have uploaded the original image of Figure 2, 3 and 4.

Comment 2: Conclusion should be revised as it concisely mentions entire findings through this study.

Response 2: Thank you very much for your suggestion. According to your guideline, we have modified the conclusion in the revised manuscript, please read Conclusions; Page 11 in the red font.

Comment 3: English throughout the text should be revised.

Response 3: Thank you very much for your great support. We have considered your comments very carefully and made the revisions in the revised manuscript. Moreover, the manuscript has been completely checked by a native English speaker. The modifications were marked in red in revised manuscript. We believe that the language in the revised manuscript has been significantly improved.

Once again, thank you very much for your comments and suggestions. We really hope that you could be satisfied with our modifications and explanations. We are also willing to make further improvements if needed and thank you again for your work and time.